# Diagnostic Pathways and Molecular Biomarkers in Colorectal Cancer: Current Evidence and Perspectives in Poland

**DOI:** 10.3390/cimb47121047

**Published:** 2025-12-15

**Authors:** Bartosz Bichalski, Magda Bichalska-Lach, Dariusz Waniczek

**Affiliations:** 1Department of Surgical Nursing and Propaedeutics of Surgery, Faculty of Health Sciences in Katowice, Medical University of Silesia, 41-902 Bytom, Poland; mbichalska@sum.edu.pl; 2Department of Oncological Surgery, Faculty of Medical Sciences in Zabrze, Medical University of Silesia, 40-514 Katowice, Poland

**Keywords:** biomarkers, colorectal neoplasms, early detection, mass screening

## Abstract

Colorectal cancer (CRC) is the third most commonly diagnosed malignancy worldwide and remains a major challenge in contemporary oncology, where early detection is critical for improving treatment outcomes and survival. Despite significant progress in diagnostics and therapy, the epidemiology, risk factors, and molecular mechanisms driving CRC development continue to be intensively investigated. This paper provides an overview of current trends in CRC diagnosis and management, with particular emphasis on advances in molecular medicine and biological sciences. Screening recommendations in Poland are discussed, comparing invasive methods—such as colonoscopy, sigmoidoscopy, and CT colonography—with non-invasive stool-based tests (FOBT, FIT, sDNA-FIT), and evaluating their sensitivity, specificity, and impact on mortality reduction. Key tumor markers with diagnostic, prognostic, and predictive value, including CEA, CA19-9, mSEPT9, ctDNA, TPS, TAG-72, CTCs, and circulating microRNAs, as well as p53 and PTEN proteins, are reviewed in the context of their clinical utility in early detection, disease monitoring, and treatment response assessment. The analysis also highlights the epidemiological situation in Poland and underscores the growing importance of integrating molecular biomarkers with traditional diagnostic methods, which may ultimately support the development of more precise and individualized clinical management strategies in the future.

## 1. Introduction

Colorectal cancer (CRC) is among the most frequently diagnosed malignancies worldwide, posing a significant challenge in prevention, diagnosis, and therapy. Risk factors for CRC include family history of the disease, inflammatory bowel diseases (such as ulcerative colitis and Crohn’s disease), diabetes, previous cholecystectomy, and postmenopausal hormone therapy. Lifestyle-related factors, such as overweight and obesity, lack of physical activity, smoking, alcohol consumption, and unhealthy dietary patterns (low intake of fiber, fruits, vegetables, calcium, and nutritional products, combined with a high intake of red and processed meat), significantly increase CRC risk. In addition, gut microbiota, age, sex, race, and socioeconomic status are known to influence the likelihood of developing CRC [1,2,3,4].

In Poland, the incidence of CRC is estimated at approximately 18,000 new cases per year. In 2021, CRC ranked as the third most commonly diagnosed cancer in both men and women. Regarding mortality, it was the second leading cause of cancer-related death in men and the third in women. Most CRC cases (65–75%) are sporadic, with age being the most significant risk factor. Approximately 10–15% of cases are familial, resulting from a combined genetic and environmental etiology.

In the remaining 5–10%, CRC is hereditary and may develop either in the context of polyposis or in the absence of increased colon polyps (Table 1).

The prognosis of CRC largely depends on the disease stage at diagnosis. In stage I, when the tumor is confined to the intestinal wall without metastases, the 5-year survival rate is approximately 90–95%. In stage II, where the cancer extends beyond the intestinal wall but lymph nodes are not involved, survival decreases to 70–85%. In stage III, characterized by regional lymph node involvement, the 5-year survival rate falls to around 50–60%. In stage IV, with distant metastases (e.g., to the liver or lungs), the survival rate drops dramatically to only 10–15% [8,9,10,11].

Treatment of CRC depends on the disease stage and the patient’s overall clinical status. In early-stage disease (stages I and II), surgical resection remains the primary therapeutic approach. In stage III, where lymph node involvement is present, adjuvant chemotherapy is administered after surgery, most commonly using regimens based on 5-fluorouracil (5-FU), oxaliplatin, or irinotecan to reduce the risk of recurrence. In advanced disease (stage IV), characterized by distant metastases, treatment may include systemic chemotherapy, surgery, and, in selected cases, radiotherapy. Radiotherapy is particularly relevant in rectal cancer and may be applied preoperatively (neoadjuvant) to downstage the tumor or postoperatively (adjuvant) to reduce the risk of local recurrence. Modern CRC management also incorporates targeted therapies and immunotherapy. Agents such as bevacizumab, an anti-angiogenic monoclonal antibody, are used in advanced disease to inhibit tumor vascularization and progression. Immunotherapy, including immune checkpoint inhibitors, is effective in patients with mismatch repair deficiency (dMMR) or high microsatellite instability (MSI-H), and offers significant benefit in a subset of patients with metastatic disease [12,13].

Early diagnosis and effective surgical and adjuvant treatment substantially improve patient outcomes; however, prognosis worsens with advancing disease stage. In selected cases, palliative care is provided to alleviate symptoms such as pain or bowel obstruction and plays an important role in maintaining quality of life. After completion of treatment, patients should undergo regular follow-up, including colonoscopy, monitoring of carcinoembryonic antigen (CEA) levels, and imaging studies such as computed tomography (CT) or magnetic resonance imaging (MRI) to facilitate early detection of recurrence.

Therefore, the aim of this review is to provide an integrated and clinically relevant overview of approaches used for the early identification of colorectal cancer, emphasizing diagnostic strategies currently implemented in Poland. This includes a comparison of invasive methods, such as colonoscopy, sigmoidoscopy, and CT colonography, with non-invasive stool-based modalities (FOBT, FIT, sDNA-FIT), highlighting their diagnostic accuracy and real-world applicability. Furthermore, the review discusses key molecular biomarkers and their prognostic, predictive, and monitoring potential, underscoring how their future integration with established diagnostic pathways may contribute to more precise and individualized clinical management within the Polish healthcare system.

## 2. Literature Search

### 2.1. Methodological Framework

This narrative review was conducted using a structured literature search across PubMed, Scopus, Web of Science, and Google Scholar. The search covered publications from 2000 to 2025, with earlier landmark studies included when historically relevant. Keywords were combined using Boolean operators and encompassed: “colorectal cancer”, “colorectal neoplasia”, “early detection strategies”, “diagnostic pathways”, “colonoscopy”, “FIT”, “stool DNA testing”, “molecular biomarkers”, “p53”, “PTEN”, “ctDNA”, “circulating tumor cells”, “liquid biopsy”, “multi-omics”, “artificial intelligence”, and “Poland”.

Articles were included if they presented diagnostic accuracy, clinical utility, prognostic value, or implementation challenges related to colorectal cancer diagnostics or molecular biomarkers. Exclusion criteria included non-English texts, abstract-only conference outputs, and papers lacking clinical relevance.

Limitations. As a narrative (non-systematic) review, this work may introduce selection bias. Heterogeneity across biomarker methodologies, assay platforms, and study populations limits comparability. Moreover, data specific to the Polish healthcare system remain limited, particularly regarding accessibility and reimbursement of molecular diagnostics.

Future directions. Further population-based research in Poland, cost-effectiveness analyses of molecular diagnostics, and prospective validation of multi-omics and AI-driven assays are warranted to support broader clinical implementation.

### 2.2. Screening for Colorectal Cancer

CRC may remain asymptomatic for many years; therefore, screening is recommended in asymptomatic individuals to enable early detection. Increased colonoscopy use in the coming years is expected to contribute to a reduction in CRC incidence and mortality.

Screening methods can be divided into direct imaging techniques and stool-based tests. Modern imaging modalities include sigmoidoscopy, computed tomography colonography, and colonoscopy (Table 2), which aim to identify structural abnormalities of the colonic wall, such as adenomas or early cancerous lesions. The estimated time from adenoma development to invasive carcinoma is approximately 10–15 years, except in specific conditions such as Lynch syndrome. Stool-based tests include fecal occult blood testing (FOBT), fecal immunochemical testing (FIT), and stool DNA testing (sDNA-FIT) (Table 3). These methods are recommended for asymptomatic individuals as part of population-based CRC screening programs [14,15].

Recommendations for CRC screening vary across countries; however, most medical societies advise initiating screening at age 50 [25,26,27,28].

**Table 3 cimb-47-01047-t003:** Comparison of stool-based screening methods [29,30,31,32,33].

Method	Mechanism of Action	Sensitivity	Specificity	Remarks
gFOBT	Detects heme peroxidase activity	19.3–44.1%	65.0–99.0%	Results may be false positives due to diet or medications.
FIT	Detects antibodies specific to human hemoglobin	79–80.4%	93.5–94%	No need for dietary restrictions, better patient acceptability, preferred method in many countries.
Mt-sDNA	Detects occult blood and genetic markers	74–86%	85–94%	High sensitivity and specificity, effective tool in CRC diagnostics.

In Poland, strategies for early identification of colorectal cancer rely predominantly on colonoscopy performed within population-targeted programs coordinated by national health authorities. Although these initiatives do not constitute an organized screening program in the classical epidemiological sense, they remain central to the early detection of precancerous lesions and early-stage cancer in asymptomatic individuals. Participation rates, however, remain suboptimal, limiting the potential effectiveness of these pathways. Non-invasive stool-based tests, while clinically valuable, are used mainly when colonoscopy cannot be performed or is contraindicated.

### 2.3. CRC Early Detection Recommendations in Poland

Polish guidelines for early detection of CRC are based on 21 scientific society recommendations, including those of the Polish Society of Clinical Oncology. The national CRC early identification, centered on colonoscopy, is dedicated to individuals aged 50–65 years who do not present symptoms such as gastrointestinal bleeding, diarrhea or constipation, unexplained weight loss, or anemia of unknown origin. Colonoscopy is also recommended for individuals aged 40–49 years with a first-degree relative diagnosed with colorectal cancer. Furthermore, the program includes individuals aged 25–49 years with a family history of Lynch syndrome, provided that the diagnosis is confirmed in a genetic counseling clinic—either by meeting the Amsterdam criteria or by genetic testing. Recommendations also applies to individuals aged 25–49 years with a family history of familial adenomatous polyposis (FAP), again requiring confirmation in a genetic counseling clinic. Within the national program, colonoscopy is available only as a first-time examination; individuals who have undergone colonoscopy within the past 10 years are not eligible. As a recommended tool, colonoscopy should be repeated every 10 years. In cases where colonoscopy is not feasible, annual FOBT is recommended. Additionally, the current “Program 40 Plus” offers free-of-charge FOBT as part of preventive care [33,34] (Table 4).

Polish diagnostic recommendations emphasize colonoscopy as the principal method for detecting colorectal neoplasia in asymptomatic adults aged 50–65 years, as well as younger individuals with hereditary risk factors. Access to diagnostic pathways is influenced by regional differences in endoscopic capacity and patient awareness, which contributes to unequal participation. Unlike several Western healthcare systems, molecular stool-based assays, including methylated DNA tests, are not routinely implemented in Poland due to infrastructural and reimbursement limitations.

Tumor markers can be classified according to various criteria, such as mechanism of origin, clinical application, or type of biological material analyzed.

By mechanism of origin in CRC, markers include:Genetic markers—mutations in specific genes, e.g., KRAS, NRAS, BRAF.Epigenetic markers—DNA methylation changes, e.g., SEPT9.Protein markers—proteins secreted by tumor cells or their microenvironment, e.g., CEA.MicroRNA (miRNA) markers—short RNA molecules regulating gene expression, e.g., miR-21.By clinical application, markers are classified as:Diagnostic markers—allow detection of an ongoing neoplastic process, e.g., methylated SEPT9 (mSEPT9), a test based on detection of methylated SEPT9 DNA in patient plasma; TAG-72; circulating miRNAs (c-miRNA); p53.Prognostic markers—help determine patient prognosis, e.g., BRAF mutation in CRC, associated with worse outcomes; TAG-72; circulating tumor cells (CTCs); circulating tumor DNA (ctDNA); c-miRNA; p53; PTEN.Predictive markers—indicate response to specific therapies, e.g., KRAS and NRAS mutations in CRC predict resistance to anti-EGFR therapy; ctDNA; PTEN.Monitoring markers—used to assess treatment effectiveness and detect recurrence, e.g., CEA, CA 19-9, TPS, TAG-72, CTCs, ctDNA, c-miRNA.By biological material analyzed, markers include:Serum markers—present in patient blood, e.g., carbohydrate antigen CEA.Tissue markers—present in surgical material, e.g., TP53 and PTEN mutations detected in paraffin-embedded tumor samples.Stool markers—present in fecal material, e.g., sDNA-FIT screening tests.Urinary markers, e.g., c-miRNA, are detectable in the early stages of CRC.

### 2.4. Selected Markers of Colorectal Cancer

Carcinoembryonic antigen (CEA) is present in many epithelial tumors. It was first identified in colorectal cancer cells, and elevated concentrations have also been observed in epithelial cells of the stomach, tongue, esophagus, cervix, and prostate, as well as in medullary thyroid carcinoma, breast cancer, and mucinous ovarian carcinoma [35]. CEA is detectable in approximately 70% of CRC patients and is primarily used in clinical practice for recurrence detection. Reported sensitivities range from 41% to 97%, and specificities from 52% to 100%. The American Society of Clinical Oncology recommends measuring CEA levels every three months for at least two years after surgery or systemic treatment. However, as a standalone test, CEA lacks sufficient sensitivity and must be complemented with other diagnostic modalities, such as computed tomography, to avoid false-negative results [36].

Carbohydrate antigen 19-9 (CA 19-9) is a glycoprotein present in CRC as well as in pancreatic and gastric cancers. Its sensitivity in CRC ranges from 26% to 48%, which limits its utility as an independent marker for therapy monitoring. Some studies suggest that, in combination with CEA, CA 19-9 may provide prognostic value in the postoperative setting. Notably, 6–22% of Caucasians are Lewis antigen–negative and therefore unable to synthesize CA 19-9. Elevated CA 19-9 levels can also occur in benign conditions such as pancreatitis, pancreatic cysts, diabetes, liver fibrosis, cholestatic diseases, and various urological, respiratory, and gynecological disorders. Current guidelines do not recommend CA 19-9 for monitoring systemic treatment response in CRC patients [37].

Methylated SEPT9 (mSEPT9) is considered a promising biomarker in CRC diagnostics. Plasma-based mSEPT9 tests demonstrate a sensitivity of approximately 73% and specificity of 87%. Meta-analysis results are comparable, with mean sensitivity of 69% (95% CI: 0.55–0.80) and specificity of 92% (95% CI: 0.89–0.95). Although less sensitive than colonoscopy, mSEPT9 provides a valuable non-invasive detection method. Preoperative mSEPT9 assessment combined with CEA improves prediction of recurrence. Interestingly, mSEPT9 appears slightly less sensitive for colon cancer (0.87) than for rectal cancer (0.93) [38,39,40,41,42].

Tissue polypeptide-specific antigen (TPS) is expressed in various malignancies, including CRC, pancreatic cancer, and salivary gland tumors. It reflects activity of specific phases of the cell cycle (S or G2) and is released following mitosis, making it a useful proliferative marker. In CRC, TPS is detectable in 60–80% of patients, with a reported specificity of 95% and sensitivity of 83% for recurrence detection [43,44,45,46,47].

Tumor-associated glycoprotein 72 (TAG-72, CA 72-4) is absent in healthy tissues and occurs in esophageal and gastric cancers. In one study, Guadagni et al. reported TAG-72 positivity in 43% of CRC patients, increasing to 60% when combined with CEA. Another study demonstrated that combining CEA, CA 19-9, and TAG-72 improved overall sensitivity to 89%. Sensitivity varies by tumor site (45.2–96.0%), and specificity ranges from 64.0% to 69.5%. TAG-72 is used in long-term oncological follow-up and CRC diagnostics [48,49,50].

Circulating tumor cells (CTCs) are associated with poor prognosis and advanced metastatic disease. Even small populations of CTCs with stem-like features may survive in the bloodstream, promoting distant metastasis. Rising CTC levels indicate disease progression, while decreasing counts may reflect treatment response. Due to their rarity (as few as 1–10 cells per mL of blood), current research focuses on improving detection methods to enable reliable clinical use [51,52].

Circulating tumor DNA (ctDNA) was initially used to assess disease stage, but its clinical utility now includes molecular profiling, evaluation of resistance mechanisms, minimal residual disease (MRD) detection, recurrence monitoring, and therapy response assessment. Elevated ctDNA levels may precede radiographic recurrence by up to 8.7 months. Clinically, ctDNA is measured at least 4 weeks after surgery to evaluate resection radicality and at least 2 weeks after systemic therapy. For long-term surveillance, ctDNA is typically monitored every 8–12 weeks [53,54].

Circulating microRNAs (c-miRNAs) offer potential for CRC prevention and early detection, as specific miRNAs are altered early in carcinogenesis. They have been detected in plasma, stool, and urine, each offering distinct advantages and limitations. Some studies report higher concentrations of tumor-derived c-miRNAs in stool compared with plasma, with stool-based assays showing higher sensitivity. Blood-based tests are inexpensive but less specific, whereas stool-derived miRNAs demonstrate higher specificity but are limited by sampling challenges. Urine-based assays are easy to collect but produce lower miRNA yields than plasma or stool testing [55].

A key aspect of CRC pathogenesis is genomic instability, which arises through three major mechanisms: chromosomal instability (CIN), microsatellite instability (MSI), and the CpG island methylator phenotype (CIMP). The majority of CRC cases develop through CIN. The multistep genetic model of CRC progression was described by Fearon and Vogelstein in 1990 (Figure 1).

The tumor suppressor protein p53, known as the “guardian of the genome,” is one of the most crucial regulators of cellular integrity. Its mutations are common in CRC and lead to impaired DNA repair and apoptosis, enabling uncontrolled tumor cell proliferation. Overall, p53 mutations are associated with worse outcomes, including shorter disease-free and overall survival. A study involving more than 3500 CRC patients confirmed the prognostic significance of p53 alterations. Dysfunctional p53 is strongly linked to resistance to conventional therapies, making it an important target for novel therapeutic strategies. Approximately 43% of CRC patients harbor TP53 mutations, which impair its tumor-suppressive activity and contribute to tumor progression. Most TP53 mutations result in loss of function, although some may lead to gain-of-function phenotypes that promote cancer stem cell survival, proliferation, invasion, and metastasis. Additional alterations include partial p53 inactivation due to mutations in regulatory genes such as ATM (13%) and DNA-PKcs (11%). Current evidence is insufficient to support the routine use of TP53 mutation status in clinical decision-making; however, some studies indicate reduced efficacy of chemotherapeutics such as 5-fluorouracil, cisplatin, temozolomide, doxorubicin, gemcitabine, and the anti-EGFR monoclonal antibody cetuximab in CRC patients with TP53 mutations [58,59,60].

Phosphatase and tensin homolog (PTEN) is a key regulator of cell-cycle progression, and its loss may confer resistance to targeted therapies such as cetuximab and panitumumab. PTEN modulates cell proliferation, invasiveness, apoptosis, migration, adhesion, and angiogenesis, and together with p53 plays a critical role in maintaining genomic stability. Loss of PTEN function occurs in approximately 20–30% of CRC cases and is more frequent in proximal than in distal tumors, reflecting distinct underlying genetic mechanisms. MSI is believed to drive tumorigenesis in right-sided CRC, whereas chromosomal instability (CIN) is more characteristic of left-sided tumors. Reduced PTEN expression correlates with larger tumor size, deeper invasion, lymph node metastasis, and poorer overall survival compared with tumors exhibiting normal PTEN expression [61,62,63,64].

In Poland, the clinical use of tumor biomarkers in colorectal cancer remains largely limited to established markers such as CEA and, occasionally, CA 19-9. Access to advanced molecular assays—including ctDNA, CTC enumeration, methylated SEPT9, and multigene next-generation sequencing panels—is restricted primarily to tertiary oncology centers. Variability in local infrastructure and lack of reimbursement further limit broader adoption. Expanding access to molecular diagnostics would enhance early detection, refine prognostic assessment, and support more personalized therapeutic decision-making.

A summary of colorectal cancer biomarkers is presented in Table 5.

## 3. Emerging Multi-Omics and AI-Driven Approaches in CRC Biomarker Discovery

Recent developments in precision oncology extend far beyond single-gene or single-marker testing. Colorectal cancer is increasingly studied using multi-omics platforms that integrate genomic, epigenomic, transcriptomic, proteomic, metabolomic, and microbiome-derived data. This systems-level approach provides a more comprehensive understanding of tumor heterogeneity and clonal evolution than any single modality alone. Integrative multi-omics analyses have identified novel prognostic signatures and potential therapeutic targets, including composite gene-expression panels, pathway-based scores, and immune–metabolic axes that stratify patients by survival and treatment response [64].

Several studies highlight the clinical potential of these strategies. Serum- and tissue-based multi-omics analyses have revealed gene and metabolite networks that correlate with CRC risk, immune infiltration, and patient outcomes, suggesting that combined molecular readouts may outperform traditional staging systems in prognostication [66]. Likewise, multi-omics models integrating circulating metabolites, inflammatory markers, and host genetic factors have helped elucidate causal pathways linking systemic metabolism with CRC susceptibility, supporting the development of composite biomarker panels rather than isolated markers [67].

Liquid biopsy technologies have also progressed from single-layer assays to multi-parameter, blood-based platforms. Integrated analysis of cfDNA methylation, mutation profiles, fragmentomics, and copy-number variation from a single blood draw enables highly accurate discrimination between CRC patients and healthy individuals and improves early detection performance. These multi-analyte approaches demonstrate how combining circulating biomarkers—such as ctDNA, epigenetic signatures, and fragmentation patterns—can yield robust diagnostic profiles suitable for population-level screening [68].

Single-cell and spatial omics further refine biomarker discovery by resolving intratumoral heterogeneity and the tumor microenvironment (TME) at cellular resolution. Single-cell RNA sequencing (scRNA-seq) and spatial transcriptomics have identified functionally distinct cell states in CRC, including immunosuppressive myeloid populations, stem-like tumor cells, and exhausted T-cell subsets, all of which correlate with prognosis and response to immunotherapy [69]. Recent studies show that specific TME-derived signatures, generated through integrated single-cell analyses, can predict outcomes in early-onset CRC and reveal potential vulnerabilities for targeted or immune-based treatments [70].

Artificial intelligence (AI) and machine learning (ML) are increasingly used to extract clinically relevant patterns from these complex datasets. Multiple ML-based models have been developed to predict CRC risk, stage, survival, and therapeutic response by integrating clinical, pathological, and molecular variables. Systematic reviews indicate that ensemble methods, neural networks, and support vector machines achieve high accuracy and strong area under the ROC curve in survival and treatment-outcome prediction, particularly when gene-expression or multi-omics features are included [71]. AI-driven models have also been applied to treatment outcome prediction in metastatic CRC, integrating staging data, laboratory findings, and genomic profiles to generate individualized risk estimates [72].

Moreover, integrative frameworks that combine multi-omics biomarkers with AI-based analytics show promise for early detection. Multi-omics liquid biopsy panels analyzed using deep learning or advanced ML classifiers have demonstrated improved sensitivity for stage I–II disease compared with single-modality assays, suggesting a potential role in population-level screening and risk-adapted surveillance [68]. However, most AI-enhanced multi-omics models remain at the research stage, with limited external validation and a scarcity of prospective clinical trials. Rigorous evaluation and standardization will be essential before these tools can be routinely adopted in clinical practice.

In Poland, multi-omics-based diagnostic platforms—including transcriptomics, epigenomics, proteomics, metabolomics, and microbiome profiling—remain available mainly within academic research environments. Routine implementation is limited by cost, lack of widespread laboratory infrastructure, and the absence of national reimbursement pathways. Artificial intelligence tools are increasingly explored in Polish academic settings; however, their integration into clinical practice remains at an early stage. Overcoming these limitations would enable more precise biological characterization of colorectal cancer and support personalized diagnostic and therapeutic strategies.

Evidence-based overview of the clinical utility of multi-omics platforms in colorectal cancer is presented in Table 6.

## 4. Challenges in Clinical Translation and Access to Molecular Diagnostics

Despite significant scientific progress, several barriers continue to hinder the widespread integration of molecular biomarkers and multi-omics assays into routine CRC care. A major challenge is the cost and reimbursement landscape for advanced molecular testing, including next-generation sequencing (NGS), ctDNA-based assays, and comprehensive multi-omics panels. Health-system reports from Europe and North America consistently demonstrate uneven access to molecular diagnostics, with substantial variability not only between countries but also among centers within the same healthcare system [74]. In many settings, reimbursement policies cover only a narrow subset of guideline-mandated tests (e.g., RAS, BRAF, MSI/MMR), limiting broader adoption of emerging biomarkers such as ctDNA-based minimal residual disease (MRD) assays or extended NGS panels.

Standardization of analytical and pre-analytical procedures represents another critical obstacle. While PCR-based MSI testing benefits from well-established consensus guidelines and validated microsatellite panels, NGS-based MSI assays and broader NGS workflows lack universally accepted standards for assay design, performance metrics, variant interpretation, and reporting [75]. Similarly, implementation studies of ctDNA testing highlight substantial heterogeneity in sample handling, sequencing depth, bioinformatic pipelines, and reporting thresholds, all of which can influence sensitivity for MRD detection and longitudinal monitoring [76]. The absence of harmonized protocols complicates inter-laboratory comparability and undermines confidence in test results across institutions.

Liquid biopsy and multi-omics platforms also require specialized infrastructure and expertise. These assays demand high-quality biobanking, rigorous quality control, and advanced bioinformatics pipelines that are not uniformly accessible outside academic or reference centers. Surveys and implementation analyses indicate that organizational and logistical constraints—such as insufficient molecular pathology capacity, limited multidisciplinary communication, and a lack of structured pathways for ordering and interpreting tests—are as restrictive as the underlying technological barriers [77]. Clinicians may additionally face difficulties in interpreting complex multi-gene reports, particularly when variants of uncertain significance or composite biomarker scores are presented without clear guideline-based recommendations.

Finally, robust clinical validation and health-economic evaluation are essential to support the routine use of new biomarkers. Although numerous studies demonstrate analytical validity and promising prognostic or predictive potential, relatively few biomarkers—especially multi-omics signatures and AI-derived models—have been evaluated in large, prospective, randomized, or real-world implementation trials assessing patient outcomes, cost-effectiveness, and clinical impact [78]. This evidence gap contributes to conservative guideline recommendations and payer reluctance to reimburse advanced assays. Overcoming these challenges will require coordinated efforts from clinicians, laboratory specialists, data scientists, regulators, and policymakers to establish standardized frameworks for assay validation, reimbursement, and integration into routine clinical workflows. Only under such conditions can molecular diagnostics and AI-assisted multi-omics platforms fully realize their potential to support personalized treatment strategies in colorectal cancer.

Poland faces several additional barriers that limit the integration of advanced molecular diagnostics into routine colorectal cancer pathways. These include restricted public reimbursement for molecular assays, uneven availability of molecular pathology laboratories, and the absence of standardized national protocols for expanded biomarker testing. Access to assays such as ctDNA-based minimal residual disease monitoring or extended NGS profiling remains limited. Addressing these constraints would require coordinated policy measures, investment in laboratory infrastructure, and incorporation of biomarker-guided pathways into national clinical recommendations.

Summary of the major barrier limitations the adoption of molecular biomarkers and multi-omics assays in routine colorectal cancer care is presented in Table 7.

## 5. Conclusions

Advances in molecular technologies and the expanding understanding of CRC genomics have reshaped current approaches to diagnosis, prognosis, and treatment. While colonoscopy and imaging remain fundamental diagnostic tools, emerging biomarkers—such as ctDNA, CTCs, and other liquid biopsy components—offer substantial potential for earlier detection and more precise monitoring of disease progression and recurrence. Integrating these molecular tools with conventional diagnostics may enable increasingly personalized therapeutic strategies.

Modern precision oncology, driven by multi-omics technologies and AI-based analytic frameworks, provides a comprehensive view of tumor heterogeneity and has revealed novel prognostic signatures and actionable molecular pathways. Innovations such as multi-omics liquid biopsy platforms, single-cell and spatial transcriptomics, and machine learning models further expand the potential for individualized risk prediction and treatment tailoring.

Despite these advances, significant barriers—limited reimbursement, lack of analytical standardization, infrastructural constraints, and insufficient prospective validation—continue to restrict the incorporation of molecular diagnostics into routine CRC care. Overcoming these challenges will require harmonized diagnostic standards, improved accessibility, and robust prospective clinical trials.

Coordinated efforts among clinicians, laboratory specialists, researchers, and policymakers will be essential to translate scientific progress into routine clinical practice. Only through such integration can molecular diagnostics and multi-omics approaches fully realize their potential to improve outcomes for patients with colorectal cancer.

In the Polish healthcare system, pathways for early identification of colorectal cancer rely predominantly on colonoscopy rather than formalized screening programs. Although effective when performed, participation remains insufficient, reducing potential population impact. Integrating emerging molecular biomarkers—including ctDNA, CTCs, methylated DNA markers, and multi-omics-based signatures—into existing diagnostic strategies may enhance precision and support more individualized management. However, widespread adoption is currently limited by reimbursement gaps, infrastructural constraints, and variable regional availability. Strengthening diagnostic pathways, expanding access to molecular technologies, and increasing public engagement represent key steps toward improving colorectal cancer outcomes in Poland.

## Figures and Tables

**Figure 1 cimb-47-01047-f001:**
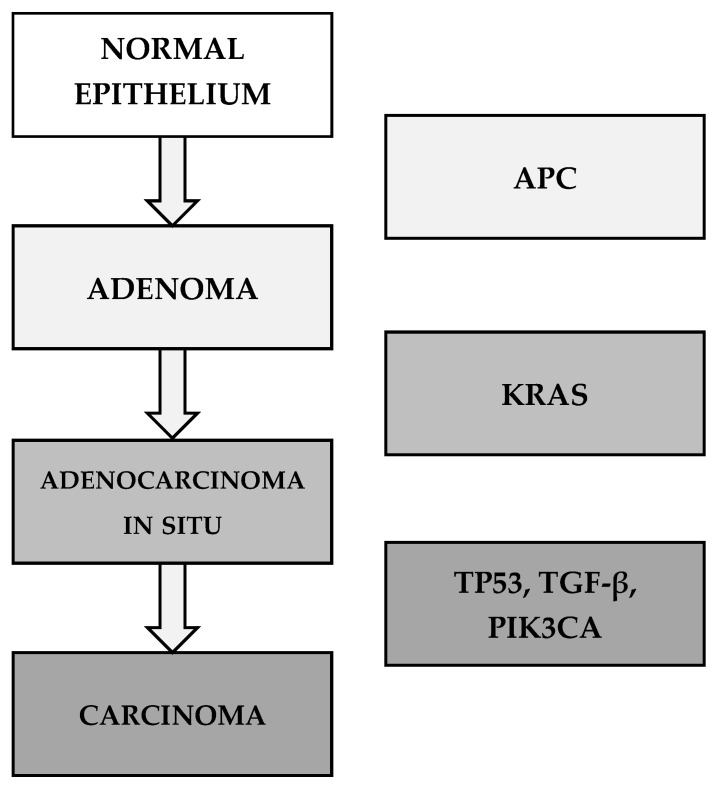
The multistep genetic model of CRC by Fearon and Vogelstein (1990) [56,57].

**Table 1 cimb-47-01047-t001:** Hereditary Colorectal Cancer Syndromes [5,6,7].

Hereditary Syndrome	Gene(s) Involved	Inheritance Pattern
FAP	*APC*	Autosomal dominant
MAP	*MUTYH*	Autosomal recessive
Peutz–Jeghers syndrome	*STK11*	Autosomal dominant
Cowden syndrome	*PTEN*	Autosomal dominant
HMPS	*CRAC1*	Autosomal dominant
Juvenile polyposis of the colon	*BMPR1A*, *SMAD4*	Autosomal dominant
HNPCC	*MLH1*, *MSH2*, *MSH6*, *PMS2*, *EPCAM*	Autosomal dominant

**Table 2 cimb-47-01047-t002:** Comparison of screening methods [16,17,18,19,20,21,22,23,24].

Method	Sensitivity	Specificity	CRC Risk/Mortality Reduction	Recommended Interval	Advantages/ Disadvantages	Refs.
Sigmoidoscopy	95% (adenomas)	87%	CRC risk ↓ 26%/mortality ↓ 30%	Every 5–10 years (with annual FIT)	Lower complication rate compared with colonoscopy, less invasive Evaluates only the distal colon	[16,17,18,19,20,21,22,23,24]
Colonography	Adenomas ≥ 10 mm: 89%; Adenomas ≥ 6 mm: 86%	Adenomas ≥ 10 mm: 94%; Adenomas ≥ 6 mm: 89%	No data on CRC mortality reduction	Every 5 years	Minimally invasive, no anesthesia, safe for patients with comorbidities, detects extracolonic abnormalities Abnormal results require colonoscopy, no evidence of mortality reduction	[16,17,18,19,20,21,22,23,24]
Colonoscopy	Gold standard; complete visualization of the colon	CRC incidence ↓ 69%	mortality ↓ up to 88%	Every 10 years (starting age: 45)	Both diagnostic and therapeutic (polyp removal), highest effectiveness More invasive, requires bowel prep and diet/medication adjustments, higher cost, complications < 1% (bleeding 0.146%, perforation 0.031%), low patient adherence	[16,17,18,19,20,21,22,23,24]

**Table 4 cimb-47-01047-t004:** Summary of colorectal cancer early detection recommendations in Poland.

Target Group	Recommended Test	Remarks	Refs.
Individuals aged 50–65 years without symptoms or with symptoms (GI bleeding, diarrhea, constipation, anemia).	Colonoscopy	First-time examination, repeated every 10 years	[33,34]
Individuals aged 40–49 years with a first-degree relative diagnosed with CRC	Colonoscopy	First-time examination, repeated every 10 years	[33,34]
Individuals aged 25–49 years with a family history of familial adenomatous polyposis (FAP)	Colonoscopy	Confirmation required in a genetic counseling clinic	[33,34]
Individuals who underwent colonoscopy within the last 10 years	Not eligible for repeat colonoscopy	Screening colonoscopy not provided under the program	[33,34]
Individuals for whom colonoscopy is not feasible	FOBT	Annual testing	[33,34]
Individuals included in the Program 40 Plus	FOBT	Free of charge under the program	[33,34]

**Table 5 cimb-47-01047-t005:** Summary of colorectal cancer biomarkers.

Biomarker	Clinical Implication	Therapeutic Consequence	Prognostic Significance	Refs.
Carcinoembryonic antigen (CEA)	Widely used marker for recurrence surveillance; elevated in ~70% of CRC.	Does not guide therapy; used for monitoring when elevated pre-treatment.	Rising postoperative CEA predicts recurrence; low sensitivity limits standalone use.	[35,36]
Carbohydrate antigen 19-9 (CA 19-9)	Low sensitivity in CRC; adjunctive marker when combined with CEA; falsely elevated in benign conditions.	Not recommended for monitoring systemic therapy or guiding treatment.	Higher levels may indicate worse outcomes; non-producers limit use.	[65]
Methylated SEPT9 (mSEPT9)	Non-invasive screening test; helpful in recurrence prediction when combined with CEA.	No direct therapeutic implications; supportive diagnostic tool.	Preoperative positivity associated with higher recurrence risk; lower sensitivity vs. colonoscopy.	[37,38,39,40,41]
Tissue polypeptide-specific antigen (TPS)	Marker of cell proliferation; useful in monitoring chemotherapy response.	Indicates treatment effect during systemic therapy.	Elevated TPS correlates with aggressive disease.	[42,43,44,45,46]
Tumor-associated glycoprotein 72 (TAG-72/CA 72-4)	Complementary diagnostic marker; increases sensitivity when combined with CEA and CA19-9.	Not used for therapy selection; adjunct for long-term follow-up.	Elevated levels associate with tumor burden; sensitivity varies by tumor site.	[47,48,49]
Circulating tumor cells (CTCs)	Reflect metastatic potential	Useful for dynamic therapy monitoring	Higher counts link to aggressive disease	[50,51]
ctDNA positivity (MRD)	Detects minimal residual disease	Guides adjuvant therapy intensity; early relapse detection	Early rise predicts recurrence months before imaging	[52,53]
Circulating microRNAs (c-miRNAs)	Early detection of CRC; stool and plasma assays complement screening; material-dependent sensitivity.	No direct therapeutic impact yet; emerging predictive potential in multi-omics models.	Altered miRNA signatures correlate with tumor stage, invasiveness, and recurrence risk.	[54]
TP53 mutation	Associated with impaired DNA damage response	Reduced sensitivity to 5-FU, cisplatin, oxaliplatin	Worse overall survival	[57,58,59]
PTEN loss	Activates PI3K/AKT pathway; induces treatment resistance	Reduced efficacy of EGFR inhibitors; potential candidate for PI3K pathway-targeted trials	Associated with deeper invasion, nodal metastases, and poor survival	[60,61,62,63]

**Table 6 cimb-47-01047-t006:** Evidence-based overview of the clinical utility of multi-omics platforms in colorectal cancer.

Approach	What It Measures	Clinical Use	Advantages	Limitations	Refs.
Genomics (NGS)	Mutations, CNVs, MSI/MMR	Therapy selection	Actionable targets	Cost; limited early detection	[64,66,67,68,69,70,71,72,73]
Epigenomics	DNA methylation	Screening; recurrence	Liquid biopsy-friendly	Lower sensitivity; variability	[64,66,67,68,69,70,71,72,73]
Transcriptomics	Gene-expression	Prognosis; immunotherapy prediction	Immune contexture captured	Needs high-quality tissue	[64,66,67,68,69,70,71,72,73]
Proteomics	Protein pathways	Target discovery	Functional insight	Low standardization	[64,66,67,68,69,70,71,72,73]
Metabolomics	Metabolic markers	Early detection	Non-invasive	High variability	[64,66,67,68,69,70,71,72,73]
Microbiome	Gut bacteria, metabolites	Risk; immune modulation	Non-invasive	Inter-individual variation	[64,66,67,68,69,70,71,72,73]
Multi-Omics Liquid Biopsy	cfDNA mutations + methylation + fragmentomics	Early detection; MRD	High accuracy	Complex analytics; cost	[64,66,67,68,69,70,71,72,73]
scRNA-seq	Single-cell TME mapping	Resistant clones; immune profiling	Highest resolution	Only research use	[64,66,67,68,69,70,71,72,73]
Spatial Transcriptomics	Spatial gene maps	TME architecture	Preserves tissue context	Very high cost	[64,66,67,68,69,70,71,72,73]
AI/ML	Integrated clinical + omics	Predicting risk/outcomes	High accuracy	Needs large datasets	[64,66,67,68,69,70,71,72,73]
Deep Learning Multi-Omics	Multi-layer biomarker fusion	Early detection	Outperforms single tests	Limited validation	[64,66,67,68,69,70,71,72,73]

**Table 7 cimb-47-01047-t007:** Summary of the major barrier limitations the adoption of molecular biomarkers and multi-omics assays in routine colorectal cancer care.

Barrier	Core Problem	Clinical Impact	Essential Solution	Refs.
Cost and Reimbursement	High cost; limited coverage	Unequal access; restricted testing	Reimbursement reform; cost-effectiveness data	[68,74,75,76,77]
Standardization Issues	Variable NGS/ctDNA workflows	Inconsistent results	Harmonized protocols; QC programs	[68,74,75,76,77]
Infrastructure Limits	Lack of sequencing/bioinformatics capacity	Limited implementation outside major centers	Investment in labs; centralized testing	[68,74,75,76,77]
Clinical Workflow Gaps	Complex reports; lack of clear pathways	Misinterpretation; underuse	Decision-support tools; MDT involvement	[68,74,75,76,77]
Evidence Gaps	Few prospective trials for new biomarkers	Conservative guidelines; low reimbursement	Large-scale validation studies	[68,74,75,76,77]

## Data Availability

No new data were created or analyzed in this study. Data sharing is not applicable to this article.

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
