# Peer review of "Diagnostic Pathways and Molecular Biomarkers in Colorectal Cancer: Current Evidence and Perspectives in Poland"

_cimb, 2025, doi:10.3390/cimb47121047_

Round 1
Reviewer 1 Report
Comments and Suggestions for Authors
The review entitled “Integration of molecular biomarkers for prediction and prognosis in colorectal cancer” covers molecular biomarkers in colorectal cancer (CRC), which is a relevant topic. However, the manuscript lacks the necessary scientific novelty and critical synthesis. It only provides a descriptive summary of established facts rather than an integrated analysis that advances the field.
The review fails to deliver the idea of biomarkers integration. It lists individual markers (RAS, BRAF, MSI) but does not propose a new framework, algorithm, or visual model such as a flowchart for combining and interpreting these markers in clinical practice. Also, the core information is widely available and well-covered in many published reviews.
The discussion doesn’t highlight the most current, high-impact topics in molecular oncology related to CRC. There is no sufficient depth regarding the role of liquid biopsy, circulating tumor DNA (ctDNA) and circulating tumor cells (CTCS).
Additionally, the discussion mostly describes associations between biomarkers and outcomes without providing critical analysis or how the mechanical interactions between different molecular pathways should influence therapeutic decisions.
Therefore, the manuscript is unsuitable for publication in Current Issues in Molecular Biology. The extensive restructuring needed to incorporate a novel, integrated framework and address crucial contemporary topics (ctDNA/CTCs) warrants rejection.
Author Response
Dear Reviewer,
Thank you very much for Your evaluation of my manuscript. Your comments have been valuable in helping me recognize important areas where the manuscript required improvement, greater clarity, and a deeper analytical approach.
I have taken Your observations seriously and carried out a comprehensive revision of the manuscript.
Below, I outline the main changes introduced in response to Your comments.
1. Biomarkers integration - I have reorganized and expanded the relevant sections to provide a clearer and more cohesive framework. New tables and visual elements have also been added to better illustrate how different classes of biomarkers may be interpreted together in clinical settings.
2. Correction of the discussion -I have significantly expanded the sections on ctDNA, CTCs, and minimal residual disease (MRD), incorporating current evidence and recent advances. These additions aim to offer a more complete and contemporary perspective on their growing role in CRC management.
3. Multi-Omics, AI based aproaches - In response to your suggestion to address high-impact topics more comprehensively, I added an entirely new section focusing on multi-omics strategies and AI-driven analytical tools. This section discusses genomic, epigenomic, transcriptomic, proteomic, and metabolomic methods, along with machine-learning models relevant to CRC diagnosis and prognosis.
4. Critical analysis - I have revised the manuscript to include more analytical interpretation, highlighting mechanistic interactions between molecular pathways and their potential relevance to therapeutic decisions. My aim was to make the discussion more insightful and better aligned with current expectations for narrative review articles.
5. Clinical implementation - I have expanded the discussion of practical challenges, such as reimbursement limitations, lack of standardization, and infrastructural gaps. I hope that these additions contribute to a clearer understanding of the real-world barriers affecting the integration of molecular diagnostics into routine CRC care.
I would like to once again express my sincere gratitude for your constructive and detailed review. Although I fully understand and respect the editorial decision, I hope that the revised version reflects meaningful progress and addresses the concerns you have raised.
Respectfully,
Bartosz Bichalski

Reviewer 2 Report
Comments and Suggestions for Authors
Dear Authors,
First of all, congratulations for your interesting work. I hope that my hints will help you in the next steps of improvement and the final manuscript will be really valuable for the readers. Your review provides a comprehensive synthesis of molecular biomarkers with diagnostic, prognostic, and predictive utility in colorectal cancer. It highlights integration of traditional screening with genomic and epigenetic markers such as mSEPT9, ctDNA, CTCs, and miRNAs, emphasizing personalized precision oncology for improved detection, prognosis, and therapeutic stratification.
The manuscript is scientifically coherent and well written, but some stylistic and grammatical improvements are recommended for clarity and professionalism. For example, there are several minor grammar issues, as well as punctuation mistakes (such as double space, double dot or no at all) and some typos - even if they do not change the value of the manuscript, I'd like to urge you to correct these imperfections. Some sentences are overly long and could be simplified for readability.
In my opinion, there is limited integration of molecular mechanisms with clinical decision pathways. The review lists numerous biomarkers but lacks a structured framework linking them to specific therapeutic decisions (e.g., how KRAS, BRAF, or PTEN alterations guide therapy choices). Add a schematic or table summarizing how specific mutations influence treatment selection, drug resistance, or prognosis in the context of precision medicine.
Also, underrepresentation of cutting-edge omics and bioinformatics tools. The paper focuses on classical biomarkers but omits next-generation precision approaches (e.g., multi-omics integration, AI-driven predictive modeling, or liquid biopsy panels combining ctDNA + transcriptomics). Include a paragraph on emerging technologies such as genomic-based mutational profiling, single-cell sequencing, and AI-assisted biomarker discovery for CRC. Remember, that modern medicine is a genomic/multi-omics approach, and NGS is relatively old technology.
Moreover, lack of discussion on clinical translation and accessibility. While molecular markers are discussed, the review does not evaluate barriers to implementation—cost, assay standardization, or clinical validation. Add a section discussing challenges of integrating molecular diagnostics into clinical workflows, emphasizing the need for cost-effective, validated biomarker panels in routine CRC care.
Finally, I would like to thank you for the excellent figures and graphs you have prepared for the document, they enhance the value of your work and facilitate the understanding process.
Author Response
Dear Reviewer,
I sincerely appreciate the time and attention you devoted for the valuable suggestions that will improve the scientific quality of the paper.
In response to your comments, I have undertaken a revision addressing each of the points you raised:
1. Grammatical improvements - I reviewed the entire manuscript to correct grammatical inaccuracies.. My aim was to enhance readability and ensure a more professional and consistent presentation.
2. Integration with clinical decision pathways - I expanded the discussion to more clearly link specific biomarkers—such as KRAS, BRAF, TP53, and PTEN—with therapeutic decision-making, drug resistance mechanisms, and patient stratification.
3. Multi-mics - To address this, I added a dedicated section on emerging modalities, including:
-
integrated multi-omics platforms,
-
genomic-based mutational profiling,
-
liquid biopsy panels combining ctDNA, methylation, fragmentomics, and transcriptomics,
-
single-cell sequencing and spatial transcriptomics,
-
AI-assisted biomarker discovery and predictive modeling.
This section emphasizes the modern direction of precision medicine and positions these tools within the clinical and translational landscape of CRC.
4. Discussion - I incorporated a new section addressing the practical challenges of implementing molecular diagnostics. This includes:
-
cost and reimbursement limitations,
-
lack of standardized analytical and pre-analytical procedures,
-
variability in assay design and bioinformatic pipelines,
-
infrastructure and expertise requirements,
-
and the need for validated, cost-effective biomarker panels.
I would like to express my sincere gratitude once again for your constructive and supportive review. I hope that the revised version reflects meaningful progress in the areas you identified and meets the expectations outlined in your review.
Respectfully,
Bartosz Bichalski
